# Preparation and Regulation of Natural Amphiphilic Zein Nanoparticles by Microfluidic Technology

**DOI:** 10.3390/foods13111730

**Published:** 2024-05-31

**Authors:** Zhe Liu, Xiaojie Ma, Yanzheng Ge, Xue Hei, Xinyu Zhang, Hui Hu, Jinjin Zhu, Benu Adhari, Qiang Wang, Aimin Shi

**Affiliations:** 1Institute of Food Science and Technology, Chinese Academy of Agricultural Sciences, Key Laboratory of Agro-Products Processing, Ministry of Agriculture and Rural, Beijing 100193, China; liuzhe019@snnu.edu.cn (Z.L.); maxiaojie@caas.cn (X.M.); heixue0022@163.com (X.H.); 82101225013@caas.cn (X.Z.); huhui@caas.cn (H.H.); zhujinjin@caas.cn (J.Z.); 2Food Laboratory of Zhongyuan, Luohe 462300, China; geyanzheng@163.com; 3College of Science, RMIT University, Melbourne, VIC 3083, Australia; benu.adhikari@rmit.edu.au; 4School of Food Science and Engineering, Nanjing University of Finance and Economics, Nanjing 210093, China; 5College of Food Science and Pharmacy, Xinjiang Agricultural University, Ürümqi 830052, China

**Keywords:** zein nanoparticles, microfluidic technology, solvent polarity, emulsifying property

## Abstract

Microfluidic technology, as a continuous and mass preparation method of nanoparticles, has attracted much attention in recent years. In this study, zein nanoparticles (ZNPs) were continuously fabricated in a highly controlled manner by combining a microfluidics platform with the antisolvent method. The impact of ethanol content (60~95%, *v*/*v*) and flow rates of inner and outer phases in the microfluidics platform on particle properties were examined. Among all ZNPS, 90%-ZNPs have the highest solubility (32.83%) and the lowest hydrophobicity (90.43), which is the reverse point of the hydrophobicity of ZNPs. Moreover, when the inner phase flow rate was 1.5 mL/h, the particle size decreased significantly from 182.81 nm to 133.13 nm as the outer phase flow rate increased from 10 mL/h to 50 mL/h. The results revealed that ethanol content had significant impacts on hydrophilic–hydrophobic properties of ZNPs. The flow rates of ethanol–water solutions and deionized water (solvent and antisolvent) in the microfluidics platform significantly affected the particle size of ZNPs. These findings demonstrated that the combined application of a microfluidics platform and an antisolvent method could be an effective pathway for precisely controlling the fabrication process of protein nanoparticles and modulating their physicochemical properties.

## 1. Introduction

With the rapid development of nanotechnology, microfluidic technology, as a precision and efficient micro and nano scale processing method, has shown unique advantages in the design and preparation of nanomaterials [1,2,3,4]. By precisely manipulating the fluid flow, mixing and reaction conditions in microfluidic channels at the micron to nanometer level, microfluidic technology achieves a high degree of control over the morphology, size, surface properties and other microscopic characteristics of nanoparticles, which greatly promotes the wide application of functional nanoparticles in biomedicine, catalysis, sensing and environmental remediation [5,6,7,8].

Among the many nanomaterials that can be regulated by microfluidic technology, natural amphiphilic zein nanoparticles have attracted extensive attention due to their excellent biocompatibility and versatility [4,9]. Zein, as a plant-derived protein, has unique hydrophobic and hydrophilic regions that enable it to self-assemble into stable nanoparticles, which not only provides an ideal carrier for drug delivery systems but also opens up new avenues for the development of green, sustainable nanomaterials [10]. At present, the research on zein nanoparticles has been gradually transitioning from basic theoretical exploration to the practical application stage, including optimizing preparation processes, regulating hydrophilic and hydrophobic, improving stability and enhancing targeting [11,12,13]. However, there are still issues with how to accurately control the size and enhance the preparation efficiency of zein nanoparticles and dynamically adjust their hydrophilic and hydrophobic interface characteristics.

It has been shown that using the microfluidic system, researchers can precisely control the mixing process of zein ethanol soluble solution with anti-solvent or other additives and effectively prepare zein nanoparticles with uniform particle size and good stability through rapid diffusion and solidification at the micro scale [8]. Against this background, microfluidic technology provides a new strategy to solve the above problems. By fine-tuning the hydrodynamic conditions, reaction time, and component ratio in microfluidic chips, zein nanoparticles with specific size and surface properties can be prepared in batches and consistently in a single operation, thus contributing to the investigation of their amphiphilic regulation.

In order to accurately control the amphiphilicity of zein nanoparticles, we used microfluidic technology to implement an anti-solvent method to prepare zein nanoparticles with size-controllable and good uniformity. The influence of the polarity of solvents used to disperse zein protein on its nanoparticle characteristics, including microstructure, particle size, hydrophilicity, hydrophobicity and emulsification, was determined and explained. The effect of inner and outer phases flow rates in the microfluidics device on particle size and its control was also determined and explained.

## 2. Materials and Methods

### 2.1. Materials

Zein (98% purity) was provided by Solaibao Co., Ltd. (Beijing, China). Ethanol (>99.7% purity) was obtained from Chemical Reagent Co., Ltd. (Beijing, China). The reagent 8-aniline-1-naphthalene sulfonic acid (ANS, >97% purity) was provided by Aladdin Reagent Co., Ltd. (Shanghai, China).

### 2.2. Analysis of Amino Acid Composition

The amino acid content of zein was evaluated using the Hitachi L-8900 amino acid analyzer (Tokyo, Japan). 35 mg zein was combined with 6 mol/L HCl (5 mL) in a Schlenk test tube (Synthware Glass V141008, Sigma-Aldrich, Louis, MI, USA). After flushing with nitrogen for 3 min, the mixture was firmly sealed, and the zein was hydrolyzed at 110 °C for 22 h. The water was then evaporated at 55 °C using a rotary evaporator, and the hydrolysate was dried in a 40 °C oven. The hydrolysate was redissolved in 1.5 mL of sodium citrate buffer (0.2 mol/L, pH 2.2) and filtered using a 0.2 µm syringe membrane. Finally, the filtrate was transferred to and analyzed with an amino acid analyzer.

### 2.3. Solubility of Zein in Ethanol

#### 2.3.1. Fluorescence Analysis

A fluorescence spectrophotometer (F-2500, Hitachi, Japan) was used to obtain the fluorescence spectra of zein dissolved in 60–95% (*v*/*v*) ethanol–water solutions. The tested wavelengths for excitation and emission were 280 nm and 250~400 nm, respectively. The scanning speed was 10 nm/s, and the slit width was 5 nm. Every test was carried out at 25.0 ± 0.5 °C.

#### 2.3.2. Ultraviolet Spectroscopic Analysis

Using a UV-2550 spectrophotometer (Shimadzu Corporation, Kyoto, Japan), the ultraviolet transmittance spectra of a 2% zein–ethanol solution dissolved in various ethanol contents (60–95%, *v*/*v*) were measured. The rate of scanning was 200 nm/min, and the range spanned was 200~900 nm.

### 2.4. Preparation of Zein Nanoparticles via Microfluidics Technology

Combining microfluidic technology and the antisolvent method, zein nanoparticles (ZNPs) were produced. To prepare a zein stock solution with a 20 mg/mL concentration of zein, the zein specimen was initially dispersed into ethanol–water solutions at 60%, 70%, 80%, 90% and 95% (*v*/*v*) contents. Next, zein stock solution was utilized as the inner phase and deionized water as the outer phase in a coaxial flow-focused chip with glass material (Yongkang Leye Company, Beijing, China). A FLOW UNIT (Fluigent, Val-de-Marne, France) regulated the flow rates of both phases, where the inner and outer phases’ flow rates were set at 1.5 mL/h and 30 mL/h, respectively, and the temperature was 25 °C, to assess the impact of ethanol contents on ZNPs. In this process, ZNPs were produced immediately, and the suspension containing the nanoparticles was gathered and freeze-dried.

### 2.5. Controlling the Particle Size of ZNPs by Microfluidics Technology

Two sets of experiments were conducted to determine that microfluidic technology was capable of controlling the size of nanoparticles. In the first test group, the inner phase’s flow rate was maintained at 1.5 mL/h, while the outer phase’s flow rates were adjusted to 10, 20, 30, 40 and 50 mL/h. The second test group’s inner phase flow rates were adjusted to 0.5, 1, 1.5, 2 and 2.5 mL/h, while the outer phase flow rate remained at 30 mL/h.

### 2.6. Characterization of ZNPs

#### 2.6.1. Microscopic Analysis

The microtopography of ZNPs produced as mentioned above was examined using a scanning electron microscope (SEM, SU8010, Hitachi, Japan) and transmission electron microscope (TEM, H-7500, Hitachi, Japan). During SEM experiments, the freshly obtained ZNP solution was placed in the middle of a clean silicon wafer and dried overnight in an oven at 40 °C. The dry silicon wafer was attached to dual-sided carbon tape and vacuum sputter-coated with gold. The image was obtained using a voltage to accelerate of 10.0 kV and magnifications of 5 K and 25 K.

During TEM examination, the freshly produced and gathered ZNPs sample solution was diluted tenfold before a drop was placed on a grid of carbon-coated copper (200 mesh). After drying at room temperature, the image was acquired at a voltage of 80 kV and a magnification of 120 K.

#### 2.6.2. Particle Size and ζ Potential

ZNPs produced as previously described were tested for particle size and ζ potential at room temperature using a Zetasizer Nano (Malvern, UK). The polydispersity index (PDI) and particle size of all samples were measured at 0.01% (*w*/*v*), whereas the ζ potential was measured at 0.1%. The protein and water had refractive indices of 1.46 and 1.33, respectively.

#### 2.6.3. Light Transmittance

The UV-2550 spectrophotometer (Shimadzu Corporation, Kyoto, Japan) was applied to measure the light transmittance spectra of ZNPs solutions made as previously reported. The test utilized a scanning range of 300~900 nm, and all measurements were conducted at 25 °C.

#### 2.6.4. Fluorescence Spectrum

The fluorescence spectra of ZNPs were obtained using the same approach as mentioned in Section 2.3.1. In this test, ZNPs concentration was 1 mg/mL.

#### 2.6.5. Fourier Transform Infrared Spectroscopy

Zein and ZNPs produced as previously mentioned were subjected to FTIR spectra measurements in the 4000~400 cm^−1^ region using an FTIR spectrometer (TENSOR 27, Saarbrucken, Germany). Peakfit software (Version 4.12, SPSS Inc., Chicago, IL, USA) was used to examine FTIR spectrum data ranging from 1600 to 1700 cm^−1^ to identify secondary structural characteristics of the protein.

#### 2.6.6. Solubility

The solubility of ZNPs in water was determined using the Lowry method [14]. The specimen was dissolved in deionized water to acquire 1% (*w*/*v*) content of ZNPs, stirred for 2 min and then centrifuged at 6000× *g* for 10 min. The protein content of the supernatant was subsequently determined using the Lowry method, which used bovine serum protein (BSA) as the reference value. The solubility was calculated as the ratio of the protein concentration within the supernatant to the original protein concentration.

#### 2.6.7. Surface Hydrophobicity

The ANS fluorescent probe method was used to assess the surface hydrophobicity (H_0_) of zein nanoparticles by the previously published method [14]. In short, ZNP solutions were made in concentrations between 0.05 and 0.5 mg/mL. Subsequently, 4 mL of ZNP solution was mixed with 20 μL of 8 mM ANS, and the mixture was kept out of the light for 10 min. Using an F-2500 fluorescence spectrometer (Hitachi, Tokyo, Japan), the fluorescence intensity was measured at 390 nm excitation wavelength and 470 nm emission wavelength. The H_0_ of the sample was determined by plotting the fluorescence intensity at the beginning of the slope against the concentration of zein.

#### 2.6.8. Total/Free Sulfhydryl Group and Disulfide Bond Contents

The total and free sulfhydryl group contents of ZNP samples were measured following Beveridge et al.’s reported method [15]. For these assays, 2 mg/mL ZNP solution was produced from each sample. To measure free sulfhydryl content, 0.5 mL ZNP solution was combined with 2.5 mL Tris-Gly buffer solution (0.09 mol/L glycine, 0.086 mol/L Tris, 0.004 mol/L EDTA, pH = 8.0). 50 μL Ellman reagent was added and shaken well to mix. This combination was at room temperature (15 min), and its absorbance was tested at 412 nm. The blank was deionized water (0.5 mL), and the rest of the method remained the same. To measure total sulfhydryl content, 0.5 mL of ZNP solution was mixed with 2.5 mL of Tris-Gly buffer solution (8 mol/L urea was added to the reagent used to test free sulfhydryl groups); otherwise, the techniques were the same as those used for estimating free sulfhydryl content. The free and total SH group contents were determined using Equation (1), whereas the disulfide bond content was computed using Equation (2).
(1)$$SH\left(\frac{\mu mol}{g}\right)=\frac{73.53\times A_{412}\times D}{C}$$
(2)$$disulfide\;bond\left(\frac{\mu mol}{g}\right)=\frac{{total}_{SH}-{free}_{SH}}{2}$$
where A_412_ was the sample absorbance at 412 nm, D was the solution’s dilution ratio and C was the zein concentration in mass (mg/mL).

#### 2.6.9. Emulsifying Properties

The emulsifying qualities were determined using Pearce and Kinsella’s turbidity method [16], which included the emulsifying activity index (EAI) and the emulsifying stability index (ESI). First, the prepared specimens were dissolved in a deionized aqueous solution (1 mg/mL) and then mixed with soybean oil in a 3:1 volume ratio. This mixture was homogenized for 1.5 min using a high-speed homogenizer at 10,000 r/min. Then, 100 μL of the emulsion was extracted from the bottom of the beaker and placed into 5 mL SDS (1 mg/mL). This mixture’s absorbance was measured at 500 nm and given the designation A_0_. That mixture was allowed to stand for 10 min. After that, 100 μL of it was added to 5 mL of SDS (1 mg/mL), and its absorbance at 500 nm was measured. The result was A_10_. Equations (3) and (4) were utilized to determine the EAI and ESI, respectively.
(3)$$EAI/\left(m^{2}/g\right)=\frac{2\times 2.303\times A_{0}\times D}{C\times \left(1-\varphi \right)\times 10^{4}}$$
(4)$$ESI\left(\%\right)=\frac{A_{10}}{A_{0}}\times 100$$
where D was the dilution ratio, *φ* was the volume fraction of soybean oil (0.25) and C was the protein mass concentration (g/mL).

### 2.7. Statistical Analysis

All experiments were conducted at least three times unless otherwise noted. These results are displayed as mean ± standard deviation. Analysis of variance was carried out with the SPSS 21 software (SPSS Inc., Chicago, IL, USA). The significant difference comparing the two mean values was determined at the 95% confidence level (*p* < 0.05). For all data analysis, Origin 9.2 software (Origin Lab Inc., Northampton, MA, USA) was utilized.

## 3. Results

### 3.1. Zein’s Amino Acid Composition

Table 1 shows the amino acid content of zein; the four primary amino acids were glutamic acid, leucine, proline and alanine, which made up 21.6%, 17.0%, 8.52% and 8.34% (*w*/*w*) of the total measured amino acids, respectively. These results were consistent with the previous report [17]. Nevertheless, this zein sample had reduced levels of arginine, methionine, lysine, histidine and arginine. 33.45% (*w*/*w*) of the total amino acids were made up of the essential amino acids, which include lysine, phenylalanine, valine, methionine, isoleucine, leucine and phenylalanine. In this zein sample, non-polar neutral amino acids accounted for 53.35% (*w*/*w*). The high amount of non-polar amino acids and the absence of alkaline amino acids revealed that it would be insoluble in water but more soluble in alkaline solutions and ethanol–water solutions with an ethanol level of 60~95% (*v*/*v*).

### 3.2. Solubility of Zein in Ethanol

#### 3.2.1. Fluorescence Spectroscopy

Fluorescence spectroscopy is used to investigate the microenvironment of fluorophores in protein solutions to gather information about their conformational changes. Figure 1A demonstrated that the highest peak of the zein solution’s endogenous fluorescence absorption spectra was about 306 nm. Protein absorption spectra are primarily concerned with the absorption values of tryptophan and tyrosine, which vary according to changes in the chromophore’s microenvironmental polarity. The fluorescence intensity at 306 nm increased considerably with increasing ethanol concentration (a decrease in solution polarity), suggesting a steady increase in the exposed zein chromophore. Kim and Xu studied how zein’s structure changed in an ethanol–water mixture. They stated that different solvent polarity alterations led to varying levels of zein polarization [18]. Furthermore, non-covalent connections created micro-polymers of different sizes between protein subunits. Likewise, the pace at which disulfide bonds form will vary depending on the intensity of the hydrophobic interactions between protein molecules [11]. In addition to protein–protein interactions, protein solution also involves protein–solvent interactions. Depending on the kind of solvent, the internal hydrophobic groups of proteins are exposed to varying degrees when they dissolve or mix with it. This eventually changes the hydrophilic/hydrophobic characteristics of protein molecules [19]. The increased fluorescence intensity suggested that the zein molecular chain had unfolded, allowing zein molecules to relax their conformation and expose additional amino acid residues, resulting in an increase in fluorescence intensity.

#### 3.2.2. Light Transmittance

Figure 1B displays the light transmittance of zein in solutions containing various percentages of ethanol. The transmittance of the 90% ethanol–water solution was the highest, while that of the 95% ethanol–water solution was the least. It has been found that as a solution’s turbidity is reduced, the strength of the optical effect and light transmittance rise. These findings suggest increased solute dissolution [20]. Thus, these zein samples exhibited the maximum solubility at 90% ethanol concentration. It was also widely known that the solubility of a solute in a solvent is determined by the solute’s characteristics, the solvent’s properties (such as polarity) and the connection between them [21]. Because of its amphiphilic nature, zein develops a micellar structure in ethanol solutions; consequently, the maximum solubility at 90% ethanol–water indicated the lowest amount of aggregation [18].

### 3.3. Physicochemical Properties of ZNPs

#### 3.3.1. Microstructure and Microscopic Morphology

Figure 2 presents the SEM images of the ZNPs generated with ethanol–water stock solutions that contained 60~95% ethanol. For the microstructure of ZNPs, there are not only discrete nanoscale particles but also aggregated/connected particles. Specifically, as the percentage of ethanol by volume increases (and the polarity of the solution decreases), the degree of particle aggregation steadily increases, which was consistent with a prior result [22]. Because of their attractive surface charge, nanoparticles frequently exhibit physical agglomeration. The “liquid-liquid dispersion” theory served as the foundation for the zein nanoparticle production in this study [23,24]. In the microfluidics platform, zein constituted an inner phase and was soluble in 60~95% ethanol by volume. When the inner component traveled at a high speed through the outer phase of deionized water, it was sheared into tiny droplets. The “coalescence” or “partial coalescence” of ethanol in scattered droplets might be compared to droplets in emulsions because of the high miscibility of ethanol and water. Zein precipitated into nanoparticles when the amount of ethanol in the distributed droplets fell below the solubilization threshold, rendering it insoluble. Shear and/or Brownian motion caused the two droplets to meet when the ethanol concentration in the scattered droplets reached a point where the zein was still soluble. The technique improved the aggregation of nanoparticles when two droplets partially or fully merged, and the zein lost its solubility prior to forming solid particles.

The dispersion of ZNPs was diluted, and the nanoparticles’ morphology was studied using TEM. Figure 3 showed that all the prepared ZNPs with sizes between 100 and 200 nm were nano-scale, where the particle size of ZNPs decreased gradually with the increase in ethanol concentration from 60% to 90%, but when the ethanol concentration was increased to 95%, the ZNP particles showed obvious aggregation, corresponding with the SEM images. The particle size of ZNPs reduces with increasing ethanol content (60~90%), which might be attributed to the creation of a larger number of nuclei and mass transfer limitations [22]. Zein molecules dispersed in the droplets solidified quickly before being shorn to a smaller size during the process of the low-ethanol content inner phase ethanol stock solution being sheared by outer phase deionized water. This was anticipated to happen in a short amount of time due to the dispersed droplets’ decreased solubility (solubility limit). Larger zein nanoparticles formed as a result, and the stock solution’s ethanol content decreased. Due to mutually attractive (such as electrostatic) forces, small sized protein nanoparticles were easily agglomerated. ZNPs produced with a 95% ethanol solution using microfluidics, in particular, exhibited the most severe aggregation. This could be attributed to the ethanol’s exposure of hydrophobic amino acids and the low solubility of protein in the inner phase. It becomes essential to control the behavior of hydrophilic amino acids when protein molecules are to be transferred from an inner, water-poor solution to an outer, water-rich solution. By lowering the exposure of hydrophobic areas, this modification controls the aggregation of protein nanoparticles.

#### 3.3.2. Particle Size and ζ Potential

The particle size distribution of ZNPs particles produced using microfluidics technology with different ethanol contents of inner phase was displayed in Figure 4A. While ZNPs produced with a 95% ethanol solution exhibited aggregation, those prepared with a 60~90% ethanol solution presented a monomodal distribution. The observations drawn from the SEM images were supported by these results. Table 2 displayed the ZNPs’ average particle sizes, which grew dramatically from 134.01 nm to 224.93 nm when the ethanol contents of the zein stock solution increased from 60% to 95%, which was consistent with prior research [21].

While the PDI of ZNPs generated with a 95% ethanol solution was 0.28, indicating a large size distribution and possibly poor dispersion stability, the PDI of ZNPs prepared with a 60~90% ethanol solution was less than 0.16, indicating a reasonably narrow size distribution. Congruent with previous findings in the literature, the data demonstrated that the polarity of the stock solution, or the variation in ethanol content, influenced the size and size distribution of nanoparticles in aqueous solutions [18,22,25]. Zein’s amphiphilic character, which processes a great ability to self-assemble, may be the cause of this phenomenon. To reduce interactions with water, the non-polar amino acids in zein could easily combine into aggregates. According to the previous report, the hydrophilic groups of zein aggregates were orientated to interact with water in a favorable way in aqueous solutions, while the hydrophobic groups were protected [26]. More of the non-polar amino acids in zein were exposed as the ethanol concentration rose [18]. It implied that the more hydrophobic portion of zein tended to combine to form big particles when the solvent (ethanol) was abruptly changed to water. The SEM pictures verified that the particle aggregation increased with the concentration of ethanol increasing. Furthermore, all ZNPs had positive ζ potential values, which could be attributed to the ionized amino acid residues that are present on their surface [25,27].

ζ potential is a measure of electrostatic and charge repulsion/attraction strength between particles, and it is also one of the basic parameters affecting stability. Similarly, in microfluidic systems, the ζ potential can be used to describe and control the electrical behavior of tiny particles such as cells and proteins. ZNPs produced with a 60~90% ethanol–water solution had a ζ potential of +38~+40 mV, which was noticeably greater than ZNPs made with a 95% ethanol solution (+34.33 mV). This finding also clarified why ZNPs aqueous dispersions prepared with 60~90% ethanol–water solution exhibited greater stability compared to those prepared with an ethanol content of 95%.

#### 3.3.3. Light Transmittance

The light transmittance of particle dispersions reflects their degree of agglomeration and solubility. It is also directly related to light absorption and the phenomenon referred to as turbidity. In general, various parameters influence the light transmittance of a nanoparticle dispersion or aggregation, including concentration, particle size and the colloid’s refractive index [28]. As observed in Figure 4B, ZNP solutions generated using microfluidics technology with varying ethanol concentrations as the inner phase exhibited considerable differences in light transmission. As the ethanol concentration increased, the light transmittance of the ZNP solutions steadily reduced. The ZNP dispersion prepared with a 60% ethanol–water solvent had the highest light transmittance, whereas the ZNP dispersion prepared with a 95% ethanol–water solvent had the lowest. According to earlier research, a decrease in light transmittance caused the turbidity of the particle dispersion to rise dramatically as the particle size decreased [25]. The microstructure and particle size data demonstrated that an uptick in ethanol content led to a progressive reduction in the size of individual particles and an increase in the degree of aggregation. In final form, this caused the aqueous dispersion’s light transmittance to fall. The PDI of the particle in the aqueous solution was likewise consistent with this fluctuation in light transmittance. With a 95% ethanol–water solvent, ZNP dispersion produced the lowest transmittance and highest PDI, indicating lower stability.

#### 3.3.4. Fluorescence Spectrum

The fluorescence spectra of ZNPs are shown in Figure 5A. It was shown that there were notable differences in the fluorescence intensity of ZNPs made with varied ethanol contents in the inner phase. All of the produced samples displayed fluorescence emission peaks close to 304 nm following excitation at 280 nm. Since tryptophan was not found in the amino acid composition, the emission peak was ascribed to the tyrosine residues (4.35%, Table 1) in zein [29]. The strength of ZNPs’ emission peak steadily reduced as the ethanol concentration increased, which suggests a decrease in solution polarity and less exposure to polar amino acid residues. The lowest emission peak was seen in ZNPs generated with a 95% ethanol–water solvent compared to other ZNPs; this could be explained by increased steric hindrance and a greater degree of particle agglomeration [30]. As demonstrated by its microstructure, this ultimately obscured the polar tyrosine and decreased fluorescence intensity. Additionally, the fluorescence intensity of ZNPs was much lower than that of zein ethanol–water solution, suggesting that the protein molecules in the solution are flexible. Using microfluidics technology, tightly structured nanoparticles were created via the antisolvent approach, and zein’s polar and non-polar amino acid positions underwent molecular reorientation.

#### 3.3.5. FTIR

FTIR delivers light on protein secondary structure and intermolecular interactions. The shifting of the absorption peak’s position suggests that something has changed about its underlying structure [31]. All of the samples, as seen in Figure 5B, had notable absorption peaks at about 3300 cm^−1^, which were likely caused by the protein molecule’s hydrogen bonds and O-H stretching vibrations [32]. As the concentration of ethanol in the inner phase rose, the absorption peak of ZNPs shifted from 3323 (zein) to 3307 (60%-ZNPs), 3306 (70%-ZNPs), 3305 (80%-ZNPs), 3305 (90%-ZNPs) and 3303 (95%-ZNPs) cm^−1^. This could be because of the different polarity of the inner phase solution and the swift exchange of the outer phase to form the hydrogen bond within the nanoparticles. Furthermore, the stretching vibration of C-H in the protein carbon skeleton was captured by the absorption peak that emerged at 2900 cm^−1^. The stretching vibrations of the C-O and C-N groups were the primary source of the protein’s characteristic peak in the amide I band (1664 cm^−1^), whereas the bending vibrations of the N-H group and the stretching vibrations of the C-N group were the main causes of the amide II band (1534 cm^−1^) [33]. All of the samples show consistency between these two bands. A common method for determining the secondary structure of proteins is to analyze the amide I band, and the results are displayed in Figure 5C. In zein, the percentages of the α-helix, β-sheet, β-turn and random coil structures are 30.04 ± 4.27, 26.16 ± 1.87, 24.00 ± 2.01 and 19.80 ± 0.98%, respectively. When subjected to microfluidics platform and anti-solvent treatment, the β-turn and random coil structures of ZNPs decreased gradually as the concentration of ethanol increased in the inner phase. Conversely, the β-sheet structure gradually increased to 28.85 (60%-ZNPs), 31.89 (70%-ZNPs), 34.83 (80%-ZNPs), 36.81 (90%-ZNPs) and 41.15% (95%-ZNPs). This increase in the β-sheet was attributed to the transformation of the disordered structure of the protein to the ordered structure [34].

### 3.4. Functional Characteristics of ZNPs

#### 3.4.1. Solubility

The solubility of proteins has a direct impact on their other physicochemical features, such as oil–water and air–water interfacial properties; consequently, increasing their solubility is critical for broadening their applications. As demonstrated in Table 3, the solubility of ZNPs increased from 16.73% (60%-ZNPs) to 32.83% (90%-ZNPs) after decreasing to 20.65% (95%-ZNPs) as the concentration of ethanol in the inner phase rose. Many factors, including particle size and aggregation, can influence the solubility of ZNPs. As can be discovered, the decreasing particle size of ZNPs assisted in improving their solubility. The reduced solubility of 95%-ZNPs was caused by agglomeration and increased exposure or interaction with hydrophobic groups to water.

#### 3.4.2. Hydrophobicity

Surface hydrophobicity (H_0_), which measures influences in the internal structure and surface microenvironment of ZNPs, is a crucial metric for assessing conformational changes and interactions; of particular importance is the exposure of hydrophobic groups [35]. Protein H_0_ is frequently assessed using the ANS fluorescent probe technique [36]. The data in Table 3 demonstrated that the H_0_ of ZNPs dropped from 187.47 (60%-ZNPs) to 90.43 (90%-ZNPs) and then climbed to 217.74 in the case of 95%-ZNPs when ethanol content increased (a reduction in solvent polarity). These findings indicated that the hydrophobic groups’ exposure on the ZNPs first dropped and then increased, which may be connected to the size of the particles, the degree of their agglomeration and the solubility of zein protein [19,37]. Particle size data and SEM images of ZNPs samples revealed that a steady drop in particle size was associated with an increase in ethanol content in the inner phase. For instance, ZNPs produced with a solvent mixture of 95% ethanol and water showed significant particle agglomeration. Furthermore, the ZNPs’ fluorescence spectrum verifies that the hydrophobic groups’ level of exposure on the sample surface varied. All of these data suggested that the microfluidics platform can assist in altering the functional features of zein and increase the efficacy of its anti-solvent treatment.

#### 3.4.3. Total Sulfhydryl/Free Sulfhydryl/Disulfide Groups

The free sulfhydryl (free-SH) groups which are visible on the protein surface and the sulfhydryl groups that are concealed inside protein molecules make up the total sulfhydryl (total-SH) groups [38]. According to the previous report, free-SH groups in proteins were easily oxidized by the processing environment to generate disulfide bonds, which were covalent connections between residues that include amino acids [39]. Disulfide connections allow protein molecules to interact with one another, altering their structure and promoting structural stability. The total-SH contents of the ZNPs were not significantly different from one another, as Table 4 illustrates, but the free-SH and disulfide bonds were significantly different. This discrepancy may have resulted from the zein protein’s ability to establish a disulfide bond, which enabled it to quickly switch between the inner and outer phases and transition from its stretched state in an ethanol solution to a nanoparticle. During this procedure, free sulfhydryl groups might have been present in the residues of sulfur-containing amino acids that were on the surface. The number of free-SH groups on the surface of the ZNPs first reduced and then increased as the inner phase’s ethanol concentration rose. In 80%-ZNPs, the disulfide bond content was highest (7.77 μmol/g), and the free-SH group content was lowest (0.90 μmol/g). This observation proved that the generation of disulfide bonds during the exchange of inner and outer phases increased the manufacture of nanoparticles when the inner phase solvent was 80% ethanol.

#### 3.4.4. Emulsifying Properties

Given that proteins include both hydrophilic and hydrophobic groups in their structures, they can adsorb at the oil–water interface, generating an interfacial film that stabilizes the emulsion [40]. The emulsifying activity index (EAI) and emulsion stability index (ESI) are used for assessing the emulsifying capabilities of proteins [41]. As seen in Table 5, the EAI of ZNPs progressively rose from 4.42 m^2^/g (60%-ZNPs) to 5.03 m^2^/g (80%-ZNPs) and then decreased to 4.66 m^2^/g (95%-ZNPs) with the rise in ethanol concentration in the inner phase (reduction of polarity). Moreover, the ESI of ZNPs increased from 49.06% (60%-ZNPs) to 61.01% (80%-ZNPs) after decreasing to 50.46% (95%-ZNPs). It is widely acknowledged that a protein’s emulsifying characteristics are influenced by its solubility, surface hydrophobicity and surface charge [37]. In this study, a microfluidics platform was employed to accurately control the solvent exchange process between the inner and outer phases, and the resulting ZNPs showed substantial changes in particle size, potential, solubility, hydrophobicity and other parameters that determined emulsifying properties.

### 3.5. Effect of Microfluidic Flow Rate on Particle Size

The impact of inner and outer phase flow rates on ZNPs was examined using an 80% ethanol–water solvent as the microfluidics inner phase. As indicated in Table 6, when the inner phase flow rate was set to 1.5 mL/h, the particle size reduced dramatically from 182.81 nm to 133.13 nm as the outer phase flow rate increased from 10 to 50 mL/h. When the outer phase flow rate was held at 30 mL/h, the particle size grew dramatically from 132.79 nm to 157.91 nm as the inner phase flow rate was raised from 0.5 to 2.5 mL/h. This result suggested that the microfluidic technology can effectively control the size of the nanoparticles, which was consistent with earlier results, because the microfluidic precisely controls the exchange process between the inner and outer phases, thus affecting the process of nucleation and the size of the nanoparticles [42].

### 3.6. Mechanism of ZNPs Regulation by Microfluidic Technology

The antisolvent method exploits the solubility differences of zein in distinct solvents, such as ethanol solutions and water, for the fabrication of nanoscale particles. Upon dissolution in an ethanol solution, zein molecules unfold flexibly, and upon exposure to an adverse solvent (water), they undergo nucleation and growth, leading to the formation of zein nanoparticles (ZNPs). As revealed by the amino acid composition analysis of zein (Section 3.1), nonpolar amino acids account for a substantial proportion of 53.35% (*w*/*w*), indicating its insolubility in water but solubility in 60 to 95% (*v*/*v*) ethanol–water solutions. The fluorescence and ultraviolet spectra of zein in ethanol solution showed that when the ethanol concentration exceeded 90%, the non-polar (hydrophobic) residues tend to be exposed externally, while at 60%~90%, they tend to be enclosed internally, resulting in the different orientation of polar and hydrophobic amino acids in corn protein at different ethanol concentrations (solvent polarity). Thus, the amphipathy of ZNPs was affected. To harness this tunability, microfluidic technology was employed to fabricate a series of ZNPs with tailored hydrophilic–hydrophobic characteristics. The systematic characterization of ZNPs prepared via microfluidics using various inner phase ethanol concentrations (Section 3.3) revealed significant differences in their physicochemical properties, particularly their microscopic structure and particle size. Notably, when the inner phase ethanol concentration increased from 60% to 90%, the solubility of the resulting ZNPs rose from 16.73% to 32.83%, subsequently decreasing to 20.65% for the 95%-ZNPs, whereas their hydrophobicity followed an inverse trend, which was related to the particle size, agglomeration degree, surface microenvironment and exposure degree of hydrophilic and hydrophobic regions of ZNPs. Furthermore, the flow rates of both the inner and outer phases in the microfluidic platform exerted a pronounced influence on the ZNPs’ size. By adjusting inner phase flow rates from 0.5 to 2.5 mL/h and outer phase flow rates from 10 to 50 mL/h, a series of ZNPs with particle sizes ranging from 133.13 to 182.81 nm could be obtained. This demonstrated that microfluidic technology enables precise control over the flow rates and mixing ratios of distinct solutions (inner phase: zein in ethanol solution; outer phase: deionized water), thereby modulating the dissolution, aggregation behavior, nucleation and growth processes of zein, ultimately regulating the hydrophilic–hydrophobic balance and particle size of the resultant ZNPs. To gain a better understanding of the regulating mechanism for ZNPs, we provide a schematic to demonstrate the formation pathway of ZNPs produced by the microfluidic platform (Figure 6).

## 4. Conclusions

We implemented a microfluidics platform to achieve precise and continuous interaction between the solvent and anti-solvent in an anti-solvent process. Five different ZNPs were fabricated by the anti-solvent process where ethanol–water solvents had a 60~95% concentration of ethanol in the inner phase of the microfluidics. The fluorescence and ultraviolet spectra of zein in these solvents showed that the ethanol concentration (polarity of the solvent) greatly affected the orientation of the polar and non-polar amino acids of the zein protein. When the ethanol content was higher than 90%, the non-polar (hydrophobic) amino acids tended to be exposed in the outer phase, and vice versa. The resulting ZNPs had significant differences in their physicochemical properties including microstructure and particle size. Furthermore, the ordered structure of zein, specifically the β-sheet conformation, gradually increased. The solubility of ZNPs increased as the ethanol content in the solvent rose from 60% to 90%, but it subsequently decreased upon reaching 95%. Simultaneously, their hydrophobicity exhibited an inverse or opposite trend. The flow rates used in the inner and outer phases of the microfluidics platform also significantly affected the particle size and helped to control the size of the ZNPs. This study provides an effective method for the accurate, continuous and controllable preparation of protein nanoparticles, which can well control particle size and amphiphilic characteristics and has broad application prospects in food and biomedical fields.

## Figures and Tables

**Figure 1 foods-13-01730-f001:**
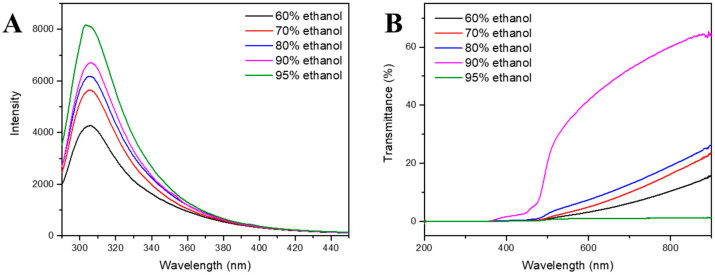
Fluorescence spectra (**A**) and ultraviolet spectroscopic (**B**) of zein dissolved in the inner phase with 60%, 70%, 80%, 90% and 95% ethanol–water solution.

**Figure 2 foods-13-01730-f002:**
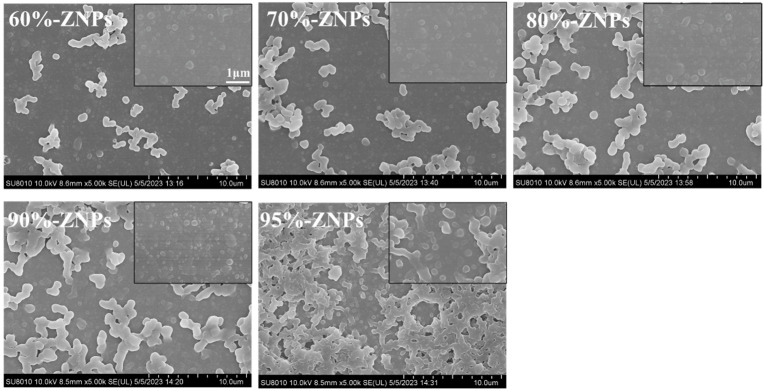
SEM images of ZNPs produced by microfluidic technology with inner phases of 60%, 70%, 80%, 90% and 95% ethanol-water solution.

**Figure 3 foods-13-01730-f003:**
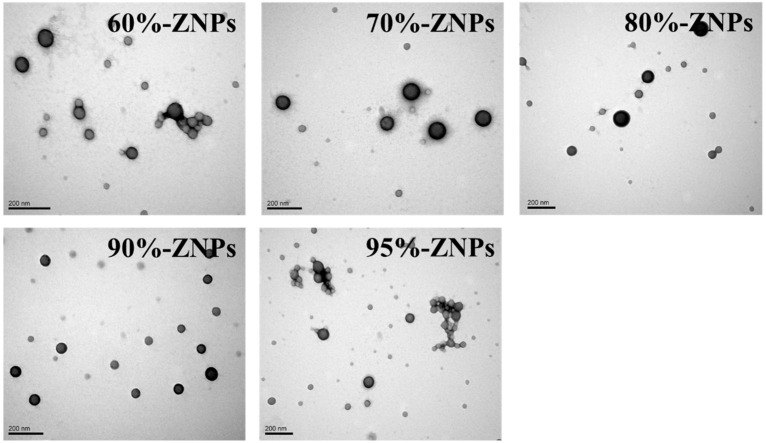
TEM images of ZNPs produced by microfluidic technology with inner phases of 60%, 70%, 80%, 90% and 95% ethanol-water solution.

**Figure 4 foods-13-01730-f004:**
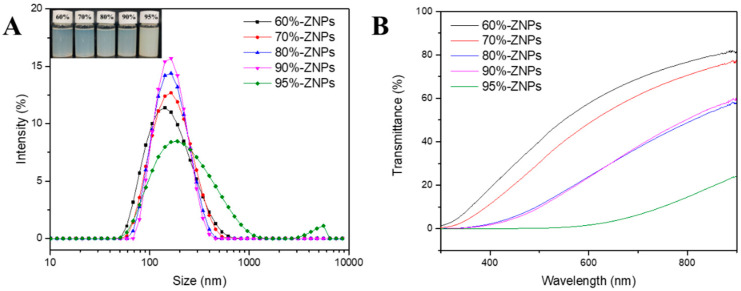
Particle size distribution (**A**), and light transmittance (**B**) of ZNPs produced by microfluidic technology with inner phases of 60%, 70%, 80%, 90% and 95% ethanol-water solution.

**Figure 5 foods-13-01730-f005:**
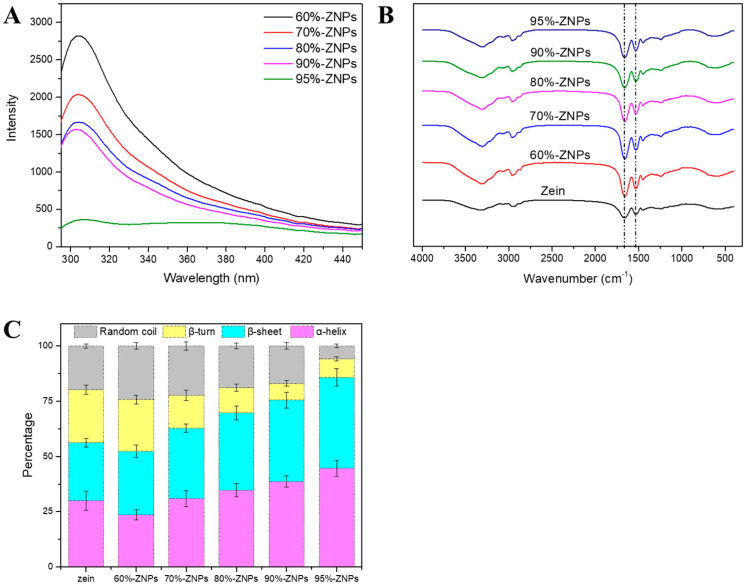
Fluorescence spectrum (**A**), FTIR (**B**) and secondary structure compositions (**C**) of ZNPs produced by microfluidic technology with inner phases of 60%, 70%, 80%, 90% and 95% ethanol–water solution.

**Figure 6 foods-13-01730-f006:**
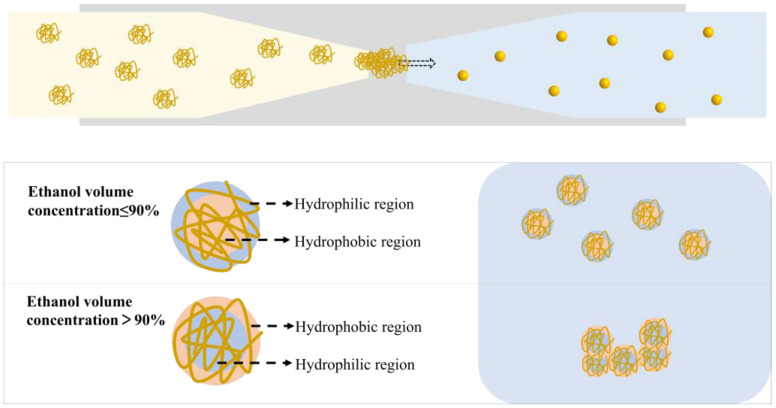
The schematic mechanism for the formation of ZNPs produced by microfluidic technology with inner phases of 60~95% ethanol–water solution.

**Table 1 foods-13-01730-t001:** Amino acid content (g/100 g protein) of native zein.

Amino Acids	Content	Amino Acid	Content
Aspartic acid	4.51 ± 0.03	Methionine	1.32 ± 0.01
Threonine	2.39 ± 0.01	Isoleucine	3.43 ± 0.02
Serine	4.53 ± 0.02	Leucine	17.00 ± 0.10
Glutamic acid	21.60 ± 0.12	Tyrosine	4.35 ± 0.03
Proline	8.52 ± 0.01	Phenylalanine	5.82 ± 0.04
Glycine	1.26 ± 0.01	Lysine	0.18 ± 0.01
Alanine	8.34 ± 0.05	Histidine	1.25 ± 0.01
Valine	3.31 ± 0.02	Arginine	1.20 ± 0.02

Data are expressed as mean ± standard deviation.

**Table 2 foods-13-01730-t002:** Mean particle size, PDI and ζ potential of ZNPs produced by microfluidic technology with inner phases of 60%, 70%, 80%, 90% and 95% ethanol–water solution.

Samples	Size (nm)	PDI	ζ Potential (mV)
60%-ZNPs	131.01 ± 2.79 d	0.15 ± 0.01 b	39.43 ± 0.95 a
70%-ZNPs	143.07 ± 1.37 c	0.16 ± 0.01 b	38.26 ± 2.64 a
80%-ZNPs	150.32 ± 1.85 b	0.14 ± 0.01 c	39.46 ± 1.82 a
90%-ZNPs	155.77 ± 1.28 b	0.13 ± 0.01 d	39.91 ± 0.78 a
95%-ZNPs	224.93 ± 1.10 a	0.28 ± 0.02 a	34.33 ± 0.42 b

Data are expressed as mean ± standard deviation, and different letters in the same column indicate significant differences (*p* < 0.05).

**Table 3 foods-13-01730-t003:** Solubility and surface hydrophobicity (H_0_) of ZNPs produced by microfluidic technology with inner phases of 60%, 70%, 80%, 90% and 95% ethanol-water solution.

Samples	Solubility (%)	H_0_
60%-ZNPs	16.73 ± 0.89 d	187.47 ± 13.2 b
70%-ZNPs	19.19 ± 1.41 c	157.75 ± 12.8 b
80%-ZNPs	23.77 ± 0.55 b	109.64 ± 9.72 c
90%-ZNPs	32.83 ± 1.03 a	90.43 ± 6.91 d
95%-ZNPs	20.65 ± 1.67 c	217.74 ± 9.93 a

Data are expressed as mean ± standard deviation, and different letters in the same column indicate significant differences (*p* < 0.05).

**Table 4 foods-13-01730-t004:** Total sulfhydryl (total-SH) group, free sulfhydryl (free-SH) group, disulfide bond of ZNPs produced by microfluidic technology with inner phases of 60%, 70%, 80%, 90% and 95% ethanol–water solution.

Samples	Total-SH Group (μmol/g)	Free-SH Group (μmol/g)	Disulfide Bond (μmol/g)
60%-ZNPs	16.65 ± 0.13 a	2.55 ± 0.14 a	7.05 ± 0.15 d
70%-ZNPs	16.46 ± 0.14 ab	1.29 ± 0.13 c	7.58 ± 0.14 b
80%-ZNPs	16.44 ± 0.23 ab	0.90 ± 0.08 d	7.77 ± 0.11 a
90%-ZNPs	16.23 ± 0.10 b	1.75 ± 0.07 b	7.23 ± 0.09 c
95%-ZNPs	16.92 ± 0.31 a	2.59 ± 0.16 a	7.15 ± 0.27 c

Data are expressed as mean ± standard deviation, and different letters in the same column indicate significant differences (*p* < 0.05).

**Table 5 foods-13-01730-t005:** Emulsifying activity index (EAI) and emulsifying stability index (ESI) of ZNPs produced by microfluidic technology with inner phases of 60%, 70%, 80%, 90% and 95% ethanol–water solution.

Samples	EAI (m^2^/g)	ESI (%)
60%-ZNPs	4.42 ± 0.64 b	49.06 ± 0.65 d
70%-ZNPs	4.47 ± 0.78 b	53.08 ± 0.84 c
80%-ZNPs	5.03 ± 0.37 a	61.01 ± 1.12 a
90%-ZNPs	4.65 ± 0.27 ab	55.50 ± 0.97 b
95%-ZNPs	4.66 ± 0.42 ab	50.46 ± 0.64 d

Data are expressed as mean ± standard deviation, and different letters in the same column indicate significant differences (*p* < 0.05).

**Table 6 foods-13-01730-t006:** The particle size of ZNPs regulated by microfluidic control.

Inner Phase Flow Rate (mL/h)	Outer Phase Flow Rate (mL/h)	Particle Size (nm)	PDI
1.5	10	182.81 ± 0.92 a	0.25 ± 0.01 a
1.5	20	169.65 ± 1.61 b	0.25 ± 0.01 a
1.5	30	148.72 ± 2.16 e	0.10 ± 0.01 d
1.5	40	140.71 ± 2.54 f	0.26 ± 0.01 a
1.5	50	133.13 ± 1.48 g	0.15 ± 0.02 b
0.5	30	132.79 ± 0.41 g	0.13 ± 0.02 c
1	30	144.92 ± 1.86 ef	0.26 ± 0.02 a
2	30	152.34 ± 0.61 d	0.13 ± 0.01 c
2.5	30	157.91 ± 2.42 c	0.14 ± 0.03 bc

Data are expressed as mean ± standard deviation, and different letters in the same column indicate significant differences (*p* < 0.05).

## Data Availability

The original contributions presented in the study are included in the article, further inquiries can be directed to the corresponding authors.

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
