# Peer review of "Preparation and Regulation of Natural Amphiphilic Zein Nanoparticles by Microfluidic Technology"

_foods, 2024, doi:10.3390/foods13111730_

Round 1

Reviewer 1 Report

Comments and Suggestions for Authors

The submitted work is processed at a very good level and corresponds to the focus of your journal. I do not have any comments on conducting the experiment. If there are already some examples of the use of zein nanoparticles in practice, I recommend mentioning them in the beginning of the article, including citations.

Author Response

Response to Reviewer 1 Comments

1. Summary

We thank the reviewers for their constructive comments on our manuscript. We have revised the manuscript according to these comments and included a detailed list of responses below. We have also highlighted the changed sections in red in the revised manuscript.

2. Point-by-point response to Comments and Suggestions for Authors

Comments 1: The submitted work is processed at a very good level and corresponds to the focus of your journal. I do not have any comments on conducting the experiment. If there are already some examples of the use of zein nanoparticles in practice, I recommend mentioning them in the beginning of the article, including citations.

Response 1: Thank you for your advice. We referenced the zein nanoparticles in the previous manuscript (Olenskyj, A.G.; Feng, Y.; Lee, Y. Continuous microfluidic production of zein nanoparticles and correlation of particle size with physical parameters determined using CFD simulation. J. Food Eng. 2017, 211, 50-59), but now have tweaked it to the beginning of the article to improve texture and readability.

Reviewer 2 Report

Comments and Suggestions for Authors

Introduction

Lines 51: Add a sentence or two summarizing key findings from earlier studies on microfluidic technology for Zein nanoparticle preparation or those involving plant-based proteins or similar materials.

Line 44:Mention any challenges that have limited its application, which your study aims to address.

 Line 61: Explicitly state the hypothesis or research question to strengthen the study's focus.

Line 61: Provide a clearer statement of the novelty of your research and its potential impact.

Materials and methods

Line 69: Ensure all materials used are specified with their purity and source

Line 98: Include specific details about the microfluidics setup such as the dimensions of the channels, the material of the device, and any unique configuration relevant to your study.

Line 100: Expand on the conditions under which the nanoparticles were fabricated, including temperature, pressure (if applicable), and any critical parameters that influence the synthesis.

Line 111: Provide specifics about the settings of the analytical instruments (e.g., SEM, TEM) such as voltage, magnification, resolution, and sample preparation methods.

Results & discussion

Particle Morphology (Lines 248-264):Include a more descriptive analysis of the SEM and TEM images, perhaps noting any observed trends or anomalies.

Lines 470-480): Clarify the more relationship between flow rates and particle size, providing a more detailed explanation of the mechanisms involved.

I like section 3.6 since it functions as a general discussion of results, but I feel results are too independent and the work could benefit of a more integrated view between the different characterizations.

Conclusions

Expand on the practical and theoretical implications of your findings. Mention how these nanoparticles could be used in specific applications or what this means for the field of nanotechnology

Line 492: Please expand on the molecular mechanisms by which ethanol concentration influences amino acid orientation and the resultant effects on nanoparticle properties.

Author Response

Response to Reviewer 2 Comments

1. Summary

We thank the reviewers for their constructive comments on our manuscript. We have revised the manuscript according to these comments and included a detailed list of responses below. We have also highlighted the changed sections in red in the revised manuscript.

2. Point-by-point response to Comments and Suggestions for Authors

Introduction

Comments 1: Lines 51: Add a sentence or two summarizing key findings from earlier studies on microfluidic technology for Zein nanoparticle preparation or those involving plant-based proteins or similar materials.

Response 1: Thanks for your good suggestion. We have added the relevant description in lines 55-58: “It has been shown that using the microfluidic system, researchers can precisely control the mixing process of zein ethanol soluble solution with anti-solvent or other additives, and effectively prepare zein nanoparticles with uniform particle size and good stability through rapid diffusion and solidification at the micro scale”.

Comments 2: Line 44: Mention any challenges that have limited its application, which your study aims to address.

Response 2: Thanks for your good suggestion. We have added the relevant description in lines 52-54: “However, there are still issues with how to accurately control size, enhance the preparation efficiency of zein nanoparticles, and dynamically adjust their hydrophilic and hydrophobic interface characteristics”.

Comments 3: Line 61: Explicitly state the hypothesis or research question to strengthen the study's focus.

Response 3: Thanks for your good suggestion. We have added the relevant description in line 64: “In order to accurately control the amphiphilicity of zein nanoparticles”.

Comments 4: Line 61: Provide a clearer statement of the novelty of your research and its potential impact.

Response 4: Thanks for your good suggestion. We have added the relevant description in lines 64-66: “We use microfluidic technology to implement an anti-solvent method to prepare zein nanoparticles with size-controllable and good uniformity”.

Materials and methods

Comments 5: Line 69: Ensure all materials used are specified with their purity and source

Response 5: The purity and source of all materials have been supplemented in lines 73-76: “Zein (98% purity) was provided by Solaibao Co., Ltd (Beijing, China). Ethanol (> 99.7% purity) was obtained from Chemical Reagent Co., Ltd (Beijing, China). The rea-gent 8-aniline-1-naphthalene sulfonic acid (ANS, > 97% purity) was provided by Alad-din Reagent Co., Ltd (Shanghai, China)”.

Comments 6: Line 98: Include specific details about the microfluidics setup such as the dimensions of the channels, the material of the device, and any unique configuration relevant to your study.

Response 6: Thank you for your suggestion. The details of the microfluidic platform, such as manufacturer and chip material, have been added in the article.

Comments 7: Line 100: Expand on the conditions under which the nanoparticles were fabricated, including temperature, pressure (if applicable), and any critical parameters that influence the synthesis.

Response 7: Thank you for your suggestion. The conditions for preparing zein nanoparticles by microfluidic technology have been added in the manuscript, mainly the inner and outer phase flow rate and temperature.

Comments 8: Line 111: Provide specifics about the settings of the analytical instruments (e.g., SEM, TEM) such as voltage, magnification, resolution, and sample preparation methods.

Response 8: Thank you for your suggestion. We have added relevant test details in 2.6.1.

Results & discussion

Comments 9: Particle Morphology (Lines 248-264): Include a more descriptive analysis of the SEM and TEM images, perhaps noting any observed trends or anomalies.

Response 9: Thank you for your suggestion. In part 3.3.1, we described the changes of particle morphology in more detail. For example: in lines 284-287: “For the microstructure of ZNPs, there are not only discrete nanoscale particles, but also aggregated/connected particles. Specifically, as the percentage of ethanol by volume increases (and the polarity of the solution decreases), the degree of particle aggregation steadily increases”; in lines 310-313: “Figure 3 showed that all the prepared ZNPs with sizes between 100 and 200 nm were nano-scale, where the particle size of ZNPs decreased gradually with the increase of ethanol concentration from 60% to 90%, but when the ethanol concentration is increased to 95%, the ZNPs particles showed obvious aggregation”.

Comments 10: Lines 470-480): Clarify the more relationship between flow rates and particle size, providing a more detailed explanation of the mechanisms involved.

Response 10: Thanks for your suggestion, in Part 3.5, lines 528-532 we re-explained the mechanism by which flow rate affects particles: “This result suggested that the microfluidic technology can effectively control the size of the nanoparticles, which was consistent with earlier results, because the microfluidic precisely controls the exchange process between the inner and outer phases, thus affecting the process nucleation and the size of the nanoparticles”.

Comments 11: I like section 3.6 since it functions as a general discussion of results, but I feel results are too independent and the work could benefit of a more integrated view between the different characterizations.

Response 11: Thank you for your suggestion. In Section 3.6, we have strengthened the correlation between the indicators. Add the corresponding statement in lines 553-582: “Notably, when the inner phase ethanol concentration is increased from 60% to 90%, the solubility of the resulting ZNPs rised from 16.73% to 32.83%, subsequently decreasing to 20.65% for the 95%-ZNPs, whereas their hydrophobicity followed an inverse trend, which was related to particle size, agglomeration degree, surface microenvironment and exposure degree of hydrophilic and hydrophobic regions of ZNPs”.

Conclusions

Comments 12: Expand on the practical and theoretical implications of your findings. Mention how these nanoparticles could be used in specific applications or what this means for the field of nanotechnology

Response 12: Thanks for your suggestion, we will re-write the practical and theoretical implications of the study in the conclusion section (lines 610-613): “This study provides an effective method for accurate, continuous and controllable preparation of protein nanoparticles, which can well control particle size and amphiphilic, and has broad application prospects in food and biomedical fields”

Comments 13: Line 492: Please expand on the molecular mechanisms by which ethanol concentration influences amino acid orientation and the resultant effects on nanoparticle properties.

Response 13: Thanks for your suggestion, we reorganized the effect of ethanol concentration on amino acid orientation and particle properties in lines 544-548: “The fluorescence and ultraviolet spectra of zein in ethanol solution showed that when the ethanol concentration exceeded 90%, the non-polar (hydrophobic) residues tend to be exposed externally, while at 60% ~ 90%, they tend to be enclosed internally, resulting in the different orientation of polar and hydrophobic amino acids in corn protein at different ethanol concentrations (solvent polarity). Thus, the amphipathy of ZNPs was affected”.

Reviewer 3 Report

Comments and Suggestions for Authors

The authors presented an interesting, well-written manuscript about the preparation and regulation of Zein nanoparticles by microfluidic technology, complying with the standards of scientific communication.

A pair of comments is provided.

In the introduction section, the authors should state a clear hypothesis of the study. Also, more background about the variables affecting the preparation of nanoparticles using microfluidic technology should be provided, including similar works supporting the possible hypothesis.

In line 87, the authors should provide a temperature range since ambient temperature varies worldwide.

In the section, the authors should discuss and explain the ζ potential with fundamentals and relate it to the possible applications and the variables assayed in microfluidic technology, as with other results.

The authors should provide a clear and concise conclusion stating the knowledge generated and answering the stated hypothesis. They should avoid the repetition of results. Also, the authors should describe the limitations of the study and the remaining questions.

Author Response

Response to Reviewer 3 Comments

1. Summary

We thank the reviewers for their constructive comments on our manuscript. We have revised the manuscript according to these comments and included a detailed list of responses below. We have also highlighted the changed sections in red in the revised manuscript.

2. Point-by-point response to Comments and Suggestions for Authors

Comments 1: The authors presented an interesting, well-written manuscript about the preparation and regulation of Zein nanoparticles by microfluidic technology, complying with the standards of scientific communication.

Response 1: Thank you for your approval. We have made point-to-point modifications to the following suggestions.

A pair of comments is provided.

Comments 2: In the introduction section, the authors should state a clear hypothesis of the study. Also, more background about the variables affecting the preparation of nanoparticles using microfluidic technology should be provided, including similar works supporting the possible hypothesis.

Response 2: Thanks for your good suggestion. According to the reviewer's suggestion, we reorganized and organized the introduction section. The example of microfluid-controlled preparation nanoparticles was supplemented in lines 55-58: ” It has been shown that using the microfluidic system, researchers can precisely control the mixing process of zein ethanol soluble solution with anti-solvent or other additives, and effectively prepare zein nanoparticles with uniform particle size and good stability through rapid diffusion and solidification at the micro scale”, and the research hypothesis of this paper is supplemented in lines 64-66: “In order to accurately control the amphiphilicity of zein nanoparticles, we use microfluidic technology to implement an anti-solvent method to prepare zein nanoparticles with size-controllable and good uniformity”.

Comments 3: In line 87, the authors should provide a temperature range since ambient temperature varies worldwide.

Response 3: Thanks for your good suggestion. We have added the corresponding temperature range "25.0 ± 0.5 °C".

Comments 4: In the section, the authors should discuss and explain the ζ potential with fundamentals and relate it to the possible applications and the variables assayed in microfluidic technology, as with other results.

Response 4: Thanks for your good suggestion. We have added the relevant description in lines 374-377: “ζ potential is a measure of electrostatic and charge repulsion/attraction strength between particles, and is also one of the basic parameters affecting stability. Similarly, in microfluidic systems, the ζ potential can be used to describe and control the electrical behavior of tiny particles such as cells and proteins”.

Comments 5: The authors should provide a clear and concise conclusion stating the knowledge generated and answering the stated hypothesis. They should avoid the repetition of results. Also, the authors should describe the limitations of the study and the remaining questions.

Response 5: Thanks for your good suggestion. We have revised the conclusion: “We implemented a microfluidics platform to achieve precise and continuous interaction between solvent and anti-solvent in an anti-solvent process. Five different ZNPs were fabricated by the anti-solvent process where ethanol-water solvents with 60 ~ 95% concentration of ethanol in the inner phase of the microfluidics. The fluorescence and ultraviolet spectra of zein in these solvents showed that the ethanol concentration (polarity of the solvent) greatly affected the orientation of the polar and non-polar amino acids of the zein protein. When the ethanol content was higher than 90%, the non-polar (hydrophobic) amino acids tended to be exposed in the outer phase, and vice versa. The resulting ZNPs had significant differences in their physicochemical properties including microstructure and particle size. Furthermore, the ordered structure of zein, specifically the β-sheet conformation, gradually increased. The solubility of ZNPs increased as the ethanol content in the solvent rose from 60% to 90%, but subsequently decreased upon reaching 95%. Simultaneously, their hydrophobicity exhibited an inverse or opposite trend. The flow rates used in the inner and outer phases of the microfluidics platform also significantly affected the particle size and helped to control the size of the ZNPs. This study provides an effective method for accurate, continuous and controllable preparation of protein nanoparticles, which can well control particle size and amphiphilic, and has broad application prospects in food and biomedical fields”.